# Role of Lipids of the Evergreen Shrub *Ephedra monosperma* in Adaptation to Low Temperature in the Cryolithozone

**DOI:** 10.3390/plants12010015

**Published:** 2022-12-20

**Authors:** Vasiliy V. Nokhsorov, Svetlana V. Senik, Valentina E. Sofronova, Ekaterina R. Kotlova, Alexander D. Misharev, Nadezhda K. Chirikova, Lyubov V. Dudareva

**Affiliations:** 1Institute for Biological Problems of Cryolithozone, Siberian Branch of Russian Academy of Sciences, 41 Lenina Av., 677000 Yakutsk, Russia; 2Komarov Botanical Institute, Russian Academy of Sciences, 2 Professor Popov str., 197376 St. Petersburg, Russia; 3Chemical Analysis and Materials Research Centre, Saint-Petersburg State University, 198504 St. Petersburg, Russia; 4Institute of Natural Science, North-Eastern Federal University, 58 Belinsky str., 677027 Yakutsk, Russia; 5Siberian Institute of Plant Physiology and Biochemistry, Siberian Branch of Russian Academy of Sciences, 132 Lermontova str., 664033 Irkutsk, Russia

**Keywords:** *Ephedra monosperma*, lipids, fatty acids, glycerolipid profiling, molecular species, seasonal dynamics, cold tolerance, phospholipids, glycolipids, adaptation, lipid metabolism

## Abstract

Lipids are the fundamental components of cell membranes and they play a significant role in their integrity and fluidity. The alteration in lipid composition of membranes has been reported to be a major response to abiotic environmental stresses. Seasonal dynamics of membrane lipids in the shoots of *Ephedra monosperma* J.G. Gmel. ex C.A. Mey. growing in natural conditions of permafrost ecosystems was studied using HPTLC, GC-MS and ESI-MS. An important role of lipid metabolism was established during the autumn-winter period when the shoots of the evergreen shrub were exposed to low positive (3.6 °C), negative (−8.3 °C) and extremely low temperatures (−38.4 °C). Maximum accumulation of phosphatidic acid (PA), the amount of which is times times greater than the sum of phosphatidylcholine and phosphatidylethanolamine (PC + PE) was noted in shoots of *E. monosperma* in the summer-autumn period. The autumn hardening period (3.6 °C) is accompanied by active biosynthesis and accumulation of membrane lipids, a decrease of saturated 34:1 PCs, 34:1 PEs and 34:1 PAs, and an increase in unsaturated long-chain 38:5 PEs, 38:6 PEs, indicating that the adaptation of *E. monosperma* occurs not at the level of lipid classes but at the level of molecular species. At a further decrease of average daily air temperature in October (−8.3 °C) a sharp decline of PA level was registered. At an extreme reduction of environmental temperature (−38.4 °C) the content of non-bilayer PE and PA increases, the level of unsaturated fatty acids (FA) rises due to the increase of C18:2(Δ9,12) and C18:3(Δ9,12,15) acids and the decrease of C16:0 acids. It is concluded that changes in lipid metabolism reflect structural and functional reorganization of cell membranes and are an integral component of the complex process of plant hardening to low temperatures, which contributes to the survival of *E. monosperma* monocotyledonous plants in the extreme conditions of the Yakutia cryolithozone.

## 1. Introduction

The adaptation of plant organisms to changing environmental conditions is one of the key problems of evolutionary biology. Plants in the process of their development grow in changing environmental conditions and are constantly exposed to various adverse factors of biotic, abiotic and anthropogenic origin. Among the variety of stress influences, one of the most significant factors is the effect of low temperatures, which makes a significant contribution to the geographical distribution of wild and agricultural plants on our planet.

In higher plants, as with other poikilothermic organisms, the most temperature-sensitive parameters are lipids of cell membranes, which provide the cell’s interaction with the external environment [1]. Membranes, in particular plasma and chloroplast membranes, are sensitive to environmental stimuli. Glycerolipids are the major constituents of membranes. In response to changes in temperature, plants can adjust the glycerolipid composition of their membranes to maintain the integrity and optimal fluidity of these membranes [2]. It was shown in the experiment with *Arabidopsis thaliana*, that after 14 days of cold acclimation at 4 °C the plants from most accessions had accumulated massive amounts of storage lipids, with most of the changes in long-chain unsaturated triacylglycerides, while the total amount of membrane lipids was changed only slightly [3]. The acclimation at the lipid level is a complex process, and the accumulation of triacylglycerides is only part of it [4]. Major changes in the relative amounts of different membrane lipids were evident. The relative abundance of several lipid species was highly correlated with the freezing tolerance of the accessions, allowing the identification of possible marker lipids for plant freezing tolerance.

The ability of the cells of frost-resistant plants to maintain the necessary membrane fluidity when the temperature regime changes is one of the important mechanisms of plant resistance to low temperatures. The main factor that allows for maintaining the fluidity of cell membranes is the effective work of cell desaturases, which provide changes in the level of unsaturated fatty acids [5]. In permafrost ecosystems of Yakutia, the maintenance of membrane fluidity is of particular biological importance, since plants are exposed to the maximum seasonal temperature amplitudes, which may reach 100 °C. Maintaining the fluidity of cell membranes, optimal for the functioning of vital proteins, is not the only mechanism of plant adaptation to the conditions of cryolithozone. The sharply continental climate probably contributed to the fact that during the long evolution plants of Yakutia have developed complex mechanisms of biochemical adaptation to extremely low temperatures, among which a significant role belongs to lipid metabolism [6].

*Ephedra monosperma* J.G. Gmel. ex C.A. Mey—a long-rooted, evergreen shrub with a height of up to 25 cm. *E. monosperma* is found in Central Yakutsk, Yana–Indigirka, Upper Lena, and Aldan floristic regions of Yakutia. This sun-loving and drought-tolerant plant grows on steppe slopes, steppificated wood edges, and on the clearings of pine forests located on calcareous rocks. Assimilation is performed by numerous branched green shoots with reduced scaly leaves.

Despite the large volume and variety of data on plant lipid metabolism, the understanding of the mechanisms of dynamic changes in lipid profiles during plant development under the influence of low and, especially, extremely low temperatures, is still limited. Therefore, the aim of this study was to find out the role of membrane glycerolipids in the evergreen shrub *E. monosperma* during adaptation to the cold climate of the Yakutian cryolithozone.

## 2. Results

### 2.1. Content of Lipid Classes

Figure 1 shows seasonal changes in polar lipids occurring in *E. monosperma* shoots during adaptation to the low temperatures of Central Yakutia (Table 1). A total of 10 classes of polar lipids were identified in *E. monosperma* shoots. Phospholipids (PL) phosphatidylcholine (PC), phosphatidylethanolamine (PE), and phosphatidic acid (PA) were the major classes in polar lipid fraction of *E. monosperma*. Phosphatidylinositol (PI), diphosphatidylglycerol (DPG), and phosphatidylglycerol (PG) were identified in smaller quantities. Glycolipids (GL) were presented by monogalactosyldiglycerid (MGDG), digalactosyldiglycerid (DGDG), and sulfoquinovosyldiacylglycerol (SQDG). Among sphingolipids glycoceramides (GlCer) were identified.

The total content of polar lipids in *E. monosperma* shoots varied depending on the season from 17.2 to 32 mg/g DW.

The most noticeable increase in the absolute content of PC and PE was found during the onset of autumn low temperatures: during September, the amount of PC in the total pool of membrane lipids increased 3.9-fold as compared to the summer value. The highest amount of PE (4.3 ± 1.1 mg/g DW) was observed when *E. monosperma* shoots were forced to rest in winter. The contribution of PA to the total pool of polar glycerolipids of *E. monosperma* shoots was maximal in summer-autumn periods, then its absolute content decreased with the onset of negative air temperatures in late October. The amount of PG increased 3.8-fold by the end of October and 4.1-fold with the onset of winter compared with the summer period. Glycolipids MGDG and DGDG were accumulated by the end of September, and then decreased under temperatures below zero.

The important indices characterizing the state of membranes in norm and under stress are PC/PE, bilayer, and non-bilayer lipids ratios (PC + PI/PE + PA + DPG), DGDG/MGDG and sum of DGDG + SQDG + PG to MGDG, shown in Figure 2. The PC to PE ratio tended to decrease smoothly from 2.3 in summer to 1.4 during the onset of low hardening temperatures in autumn period and up to 1.1 in winter period. The predominance of non-bilayer over bilayer lipids in the autumn and winter periods is illustrated by PC + PI/PE + PA + DPG ratio. The ratios DGDG/MGDG and all “bilayer” chloroplast lipids to MGDG (DGDG + SQDG + PG)/MGDG demonstrated opposite seasonal dynamics that is a slight decrease in September during hardening with low positive autumn temperatures and an increase under sub-zero temperatures.

### 2.2. Fatty-Acid Composition of Lipids

The values of seasonal changes in the fatty acid composition of lipids in shoots of the evergreen shrub *E. monosperma* are presented in Table 2. GC-MS analysis of FA methyl esters (FAMEs) revealed a fairly large variety of both saturated and unsaturated: mono-, di-, tri-, and tetraene FAs. Of saturated FAs, palmitic acid C16:0 dominated in all seasons. Of the unsaturated monoene FAs, C16:1(Δ7), C16:1(Δ9), C16:1(Δ11), cis-vaccenic acid C18:1(11t), and C18:1(Δ9) were identified. Of the polyunsaturated fatty acids (PUFA), linoleic C18:2(Δ9,12) and α-linolenic C18:3(Δ9,12,15) acids dominated in all seasons. ∆5-unsaturated polymethylene-interrupted FAs (Δ5-UPIFA): taxoleic-C18:2(Δ5,9), coniferonic- C18:4(Δ5,9,12,15) and juniperonic-C20:4(Δ5,11,14,17) were also found to be present in fairly large amounts in *Pinaceae* seed lipids [7].

In September (phase I of cold hardening), total FA unsaturation raised by increasing the proportion of C18:2(Δ9.12) and decreasing the proportion of C16:0. In October (phase II of cold hardening) the level of unsaturation continued to increase due to the accumulation of C18:2(Δ9,12) and C18:3(Δ9,12,15) acids and maintained at the same level in winter.

Seasonal changes in the absolute and relative FA content of total lipids and the calculated activities of the corresponding desaturases were reflected in the changes in the desaturation ratios, more precisely, in stearoyl desaturation ratio (SDR), oleoyl desaturation ratio (ODR), and linoleoyl desaturation ratio (LDR) (Table 2).

### 2.3. Lipid Profiling

The analysis of lipid extracts of *E. monosperma* by TOF-MS after the separation of individual lipid classes by HPTLC allowed to identify 101 molecular species. Phospholipids PC, PE, and PA, consisted of 16–18 molecular species with 34:2, 36:4, and 36:3 as the dominant molecular species (Figure 3). PE was the most heterogeneous class in terms of molecular species composition. Glycolipids MDGD and DGDG included 10–13 molecular species with the degree of unsaturation ranging from 0 to 7 (Figure 4). The most abundant species in the class were the 36:6 MGDG at *m*/*z* 797.52 and 36:6 DGDG at *m*/*z* 959.58. Sulfolipid SQDG included 12 molecular species, the most abundant ones were 34:3 and 34:2. PG consisted of 16 molecular species with 34:2, 34:1, and 34:3 as dominant molecular species.

Lipid profiling allowed to analyze seasonal changes in lipid molecular species. The levels of polyunsaturated lipid species of phospholipids such as 36:5 PC, 36:4 PC, 36:5 PE, 36:4 PE, 34:3 PE, 36:5 PA, 36:4 PA, and 36:3 PA, increased during cold acclimation (stage 1 in September and stage 2 in October) and maintained on the level of cold-acclimated plants during winter period. Monounsaturated lipid species 34:1 drastically decreased in September in all studied phospholipids. The diunsaturated molecular species of lipids tend to increase in September and decrease in October, apparently, being a substrate for the formation of more unsaturated lipids. The content of long-chain (40:2) molecular species of PL increased by winter.

The plastidic lipids MGDG and DGDG included 10–13 molecular species with the degree of unsaturation ranging from from 0 to 7. The most abundant species in each class were 36:6 MGDG at *m*/*z* 797.52 and 36:6 DGDG at *m*/*z* 959.58. Sulfolipid SQDG included 12 molecular species, the most abundant ones were 34:3 SQDG and 34:2 SQDG. The only phospholipid species of thylakoid membranes PG consisted of 16 molecular species with 34:2 PG, 34:1, and 34:3 being the dominant molecular species.

Plastidic lipids demonstrated similarity to phospholipids in seasonal dynamics: an increase in the polyunsaturated 36:6 MGDG and 36:6 DGDG during cold acclimation (stage 1 in September and stage 2 in October) and decreased of di- and tri-unsaturated molecular species. SQDG and PG are more suturated, with 34:3 SQDG and 34:2 PG molecular species accumulating during cold acclimation.

In the autumn-winter season, *E. monosperma* contained higher amounts of 34:2 PG (more than 50%), and summer extracts contained 45% of 34:1 PG and smaller amounts of 34:2 PG (32%).

Seasonal dynamics of such parameters of membrane lipids as double bond index (DBI) and acyl chain length (ACL) are depicted on the Table 3 and Table 4.

### 2.4. Figures, Tables

**Figure 1 plants-12-00015-f001:**
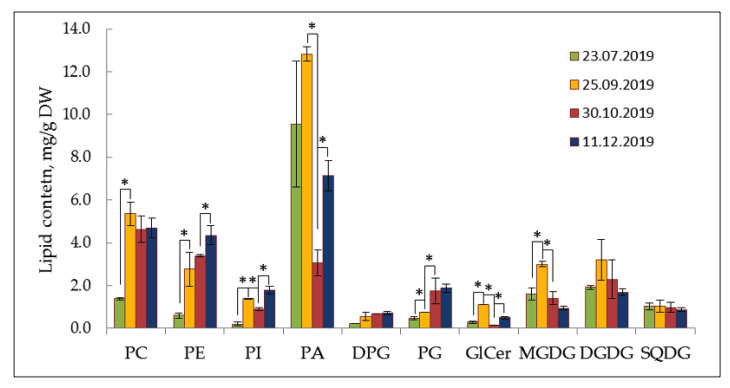
Seasonal content of each class of major membrane lipid in *Ephedra monosperma* (mg/g DW). PC—phosphatidylcholine; PI—phosphatidylinositol; PE—phosphatidylethanolamine; PG—phosphatidylglycerol; PA—phosphatidic acid; DPG—diphosphatidylglycerol; GlCer—glycoceramide, DGDG—digalactosyldiglyceride, MGDG—monogalactosyldiglyceride, SQDG—sulfoquinovosyldiacylglycerol. Data are shown as the mean ± SD (*n* = 3); * significant differences according to Student’s *t*-test.

**Figure 2 plants-12-00015-f002:**
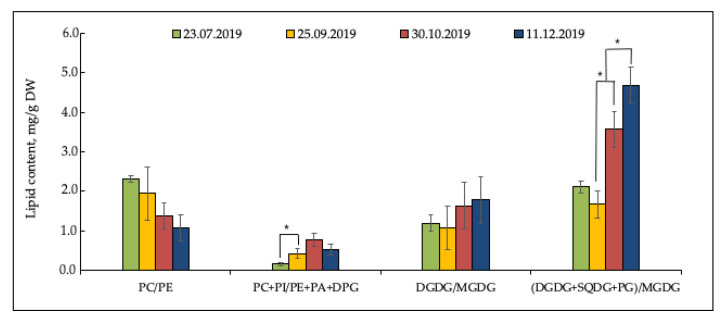
Dynamics of PC/PE, PC + PI/PA + PA + DPG, DGDG/MGDG, (DGDG + SQDG + PG)/MGDG ratios in *Ephedra monosperma* shoots (mg/g DW). PC—phosphatidylcholine; PE—phosphatidylethanolamine; PI—phosphatidylinositol; PA—phosphatidic acid; DPG—diphosphatidylglycerol; PG—phosphatidylglycerol; DGDG—digalactosyldiglycerid, MGDG—monogalactosyldiglycerid, SQDG—sulfoquinovosyldiacylglycerol. Data is shown as the mean ± SD (*n* = 3); * significant differences according to Student’s *t*-test.

**Table 1 plants-12-00015-t001:** Timing of developmental stages for *E. monosperma* plants and meteorological parameters during the research period (Central Yakutia).

Sampling Date	Growth Stage	Daily Average Air Temperature, °C ^1^	Total Precipitation mm ^2^	Photoperiod, h
2019				
12.07	Seed maturity	23.9 ± 5.8	9.7	19.0
23.07	Seed maturity	17.1 ± 1.5	9.6	17.5
15.08	Bud setting	14.9 ± 2.2	6.0	16.0
11.09	Onset of stage 1 hardening	7.2 ± 1.6	43.6	13.4
24.09	Stage 1 hardening	5.1 ± 2.4	0.5	12.1
30.09	Onset of stage 2 hardening	−0.2 ± 1.3	0.5	12.1
07.10	Stage 2 hardening	−1.1 ± 2.6	0.5	11.0
30.10	Completion of stage 2 hardening	−8.3 ± 1.6	(1) wet snow	8.7
11.12	Enforced dormancy	−38.4 ± 1.1	(18)	5.3
2020				
21.01	Enforced dormancy	−48.4 ± 0.8	(27)	6.6

^1^—Average data for 48 h prior to sampling; ^2^—the amount within 10 days before sampling, the depth of snow cover (cm) is presented in parenthesis. Data is taken from the websites http://meteocenter.net/24959_fact.htm (accessed on 17 September 2022), http://timezone.ru/suncalc.php?tid=74 (accessed on 17 September 2022).

**Table 2 plants-12-00015-t002:** Seasonal dynamics of fatty acyl content of lipids in shoots of *E. monosperma* growing in Central Yakutia.

Fatty Acids	23 July 2019 (23.9 °C ^1^)		25 September 2019 (3.6 °C ^1^)		30 October 2019 (−8.3 °C ^1^)		11 December 2019 (−38.4 °C ^1^)	
	µg/g DW	%	µg/g DW	%	µg/g DW	%	µg/g DW	%
C14:0	70.1 ± 4.0 ^a^	0.8 ± 0.0 ^a^	62.9 ± 12.1 ^a^	0.6 ± 0.0 ^a^	64.9 ± 2.5 ^a^	0.4 ± 0.0 ^a^	74.4 ± 8.5 ^a^	0.5 ± 0.0 ^a^
C15:0	32.2 ± 1.1 ^a^	0.4 ± 0.0 ^a^	30.5 ± 4.9 ^a^	0.3 ± 0.0 ^a^	32.1 ± 0.3 ^a^	0.2 ± 0.0 ^a^	29.9 ± 1.8 ^a^	0.2 ± 0.0 ^a^
C16:0	1992.6 ± 87.6 ^a^	22.6 ± 0.1 ^a^	1944 ± 490.9 ^a^	19.1 ± 2.4 ^a^	2637.2 ± 161.0 ^a^	16.7 ± 0.9 ^a^	2646.8 ± 188.9 ^a^	17.7 ± 0.3 ^a^
C16:1(Δ7)	53.0 ± 2.5 ^a^	0.6 ± 0.0 ^a^	54.2 ^a^ ± 6.4 ^ab^	0.5 ± 0.0 ^ab^	68.9 ^b^ ± 3.3 ^b^	0.4 ± 0.0 ^b^	82.3 ^b^ ± 3.5 ^b^	0.5 ± 0.0 ^b^
C16:1(Δ9)	29.4 ± 6.1 ^a^	0.3 ± 0.1 ^a^	46.8 ± 0.6 ^ab^	0.5 ± 0.1 ^ab^	74.6 ± 1.5 ^ab^	0.5 ± 0.0 ^ab^	82.2 ± 0.7 ^b^	0.5 ± 0.0 ^b^
C16:1(Δ11)	18 ± 1.1 ^a^	0.2 ± 0.0 ^a^	11.3 ± 0.7 ^a^	0.1 ± 0.0 ^a^	30.1 ± 0.2 ^a^	0.2 ± 0.0 ^a^	27.1 ± 0.3 ^a^	0.2 ± 0.0 ^a^
C17:0	22.4 ± 0.2 ^a^	0.3 ± 0.0 ^a^	36.8 ± 2.2 ^ab^	0.4 ± 0.0 ^ab^	47.9 ± 2.0 ^ab^	0.3 ± 0.0 ^ab^	53.8 ± 7.1 ^b^	0.4 ± 0.0 ^b^
C18:0	466.9 ± 25.1 ^a^	5.3 ± 0.0 ^a^	534.2 ± 152 ^a^	5.2 ± 0.8 ^a^	532.0 ± 51.1 ^a^	3.4 ± 0.3 ^a^	699.7 ± 94.9 ^a^	4.7 ± 0.4 ^a^
C18:1(Δ9)	679.4 ± 33.8 ^a^	7.7 ± 0.0 ^a^	611.9 ± 186.5 ^a^	6.0 ± 0.9 ^b^	861.1 ± 76.2 ^a^	5.4 ± 0.4 ^a^	938.6 ± 168.2 ^a^	6.2 ± 0.8 ^a^
C18:1(Δ11t)	196.1 ± 1.6 ^b^	2.2 ± 0.1 ^a^	453.5 ± 50.8 ^a^	4.5 ± 0.1 ^a^	819.7 ± 30.1 ^a^	5.2 ± 0.2 ^a^	726.3 ± 48.2 ^a^	4.8 ± 0.1 ^a^
C18:2(Δ5,9)	35.5 ± 2.9 ^a^	0.4 ± 0.0 ^a^	51.6 ± 4.8 ^a^	0.5 ± 0.0 ^a^	168.1 ± 15.8 ^a^	1.1 ± 0.1 ^a^	93.2 ± 11.6 ^a^	0.6 ± 0.1 ^a^
C18:2(Δ9,12)	1638.0 ± 105.7 ^b^	18.6 ± 0.3 ^a^	2394 ± 156.8 ^ab^	23.8 ± 1.5 ^a^	4252.1 ± 43.6 ^a^	26.9 ± 0.5 ^a^	3941.7 ± 121.5 ^ab^	26.3 ± 0.6 ^a^
NI	7.2 ± 0.7 ^a^	0.1 ± 0.0 ^a^	16.5 ± 0.4 ^a^	0.2 ± 0.0 ^a^	39.2 ± 3 ^a^	0.2 ± 0.0 ^a^	19.6 ± 2.3 ^a^	0.1 ± 0.0 ^a^
C18:3(Δ9,12,15)	2383.3 ± 213.1 ^a^	27 ± 1.1 ^a^	2264.9 ± 38.8 ^ab^	22.6 ± 2.5 ^ab^	4205.4 ± 180.2 ^b^	25.6 ± 1.3 ^b^	3769.8 ± 48.9 ^ab^	25.2 ± 1.0 ^ab^
C18:4(Δ5,9,12,15)	20.9 ± 3.2 ^a^	0.2 ± 0.0 ^a^	37.6 ± 3.1 ^a^	0.4 ± 0.0 ^a^	56.6 ± 4.8 ^a^	0.4 ± 0.0 ^a^	46.4 ± 5 ^a^	0.3 ± 0.0 ^a^
C20:0	287 ± 41.8 ^a^	3.3 ± 0.7 ^a^	303.1 ± 29 ^a^	3 ± 0.1 ^a^	253.3 ± 7.4 ^a^	1.6 ± 0.1 ^a^	218.2 ± 10.1 ^a^	1.5 ± 0.0 ^a^
C20:3(Δ7,11,14)	302.9 ± 14.7 ^a^	3.4 ± 0.0 ^a^	598.6 ± 51.6 ^ab^	5.9 ± 0.2 ^ab^	723 ± 38.1 ^b^	4.6 ± 0.3 ^b^	681.6 ± 53.3 ^ab^	4.5 ± 0.1 ^ab^
C20:4(Δ5,11,14,17)	98.1 ± 0.8 ^a^	1.1 ± 0.0 ^a^	28.3 ± 1.5 ^ab^	0.3 ± 0.0 ^ab^	51.1 ± 4.6 ^ab^	0.3 ± 0.0 ^ab^	111 ± 1.7 ^b^	0.7 ± 0.0 ^b^
C22:0	472.0 ± 19.2 ^a^	5.4 ± 0.5 ^a^	609.8 ± 109.4 ^ab^	6.0 ± 0.3 ^ab^	846.7 ± 44.8 ^b^	5.4 ± 0.2 ^b^	733.9 ± 54.9 ^ab^	4.9 ± 0.1 ^ab^
C23:0	–	–	–	–	67.4 ± 1.3	0.4 ± 0.0	–	–
∑_saturated_	3343.1 ± 54.3 ^a^	38 ± 1.3 ^a^	3521.4 ± 800.5 ^a^	34.7 ± 3.5 ^a^	4481.5 ± 244 ^a^	28.3 ± 1.3 ^a^	4456.6 ± 366.3 ^b^	29.7 ± 0.8 ^b^
∑_unsaturated_	5461.8± 384.8 ^a^	62 ± 1.3 ^a^	6569.2 ± 484 ^ab^	65.3 ± 3.5 ^a^	11,350 ± 125.9 ^ab^	71.7 ± 1.3 ^a^	10,519.9 ± 442 ^b^	70.3 ± 0.8 ^a^
SDR		0.6		0.5		0.6		0.6
ODR		0.9		0.9		0.9		0.9
LDR		0.6		0.5		0.5		0.5
DBI		1.5		1.5		1.6		1.6

Note: “–”—acid not found; ^1^—mean daily air temperature; NI—not identified fatty acid, presumably C18:3 isomer; Σsaturated—the sum of saturated FAs; Σunsaturated—the sum of unsaturated FAs; DBI—FA double-bond index; ODR—oleoyl desaturation ratio; SDR—stearoyl desaturation ratio; LDR—linoleoyl desaturation ratio. The table shows the average values from 3–6 biological replicates and their standard deviations. The significance of differences between the compared mean values was assessed Kruskal-Wallis ANOVA by ranks (*p* < 0.05). Different superscript letters indicate significant differences of analyzed parameters.

**Figure 3 plants-12-00015-f003:**
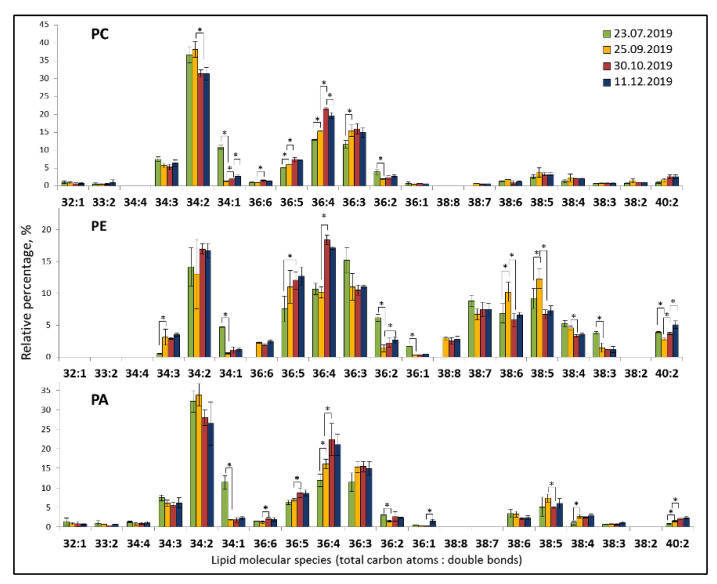
Seasonal changes in extraplastidic lipid molecular species in *Ephedra monosperma*. PC—phosphatidylcholines; PE—phosphatidylethanolamines; PA—phosphatidic acids. Bars represents standard deviations. *—significant differences according to Student’s *t*-test.

**Table 3 plants-12-00015-t003:** Double-bond index (DBI) of membrane lipids of *Ephedra monosperma*.

	23 July 2019 (23.9 °C)	25 September 2019 (3.6 °C)	30 October 2019 (−8.3 °C)	11 December 2019 (−38.4 °C)
PC	2.7 ± 0.03	3.0 ± 0.1	3.1 ± 0.03 *	3.1 ± 0.02
PE	3.6 ± 0.02	4.0 ± 0.01 *	3.9 ± 0.2	4.1 ± 0.1
PA	2.9 ± 0.2	3.2 ± 0.1	3.3 ± 0.1	3.3 ± 0.2
MGDG	5.5 ± 0.02	5.5 ± 0.03	5.6 ± 0.01	5.6 ± 0.02
DGDG	4.6 ± 0.08	5.0 ± 0.04	5.2 ± 0.03 *	5.2 ± 0.02
SQDG	2.5 ± 0.03	2.4 ± 0.02	3.0 ± 0.03 *	3.0 ± 0.02
PG	1.6 ± 0.02	1.8 ± 0.02 *	1.9 ± 0.01	1.9 ± 0.02

Note: DBI = (∑[N × % lipid])/100, N is the total number of double bonds in the two fatty acid chains of each glycerolipid molecule. PC—phosphatidylcholines; PE—phosphatidylethanolamines; PG—phosphatidylglyceroles; PA—phosphatidic acids; DGDG—digalactosyldiglycerides, MGDG—monogalactosyldiglycerides, SQDG—sulfoquinovosyldiacylglyceroles. The table shows the average values and the standard deviations. *—significant differences according to Student’s *t*-test.

**Table 4 plants-12-00015-t004:** Acyl chain length (ACL) of membrane lipids of *Ephedra monosperma*.

	23 July 2019 (23.9 °C)	25 September 2019 (3.6 °C)	30 October 2019 (−8.3 °C)	11 December 2019 (−38.4 °C)
PC	34.6 ± 0.16	34.2 ± 0.9	34.5 ± 0.1	34.4 ± 0.3
PE	35.7 ± 1.0	33.3 ± 0.8 *	34.2 ± 1.0	35.8 ± 0.01
PA	35.2 ± 0.3	35.4 ± 0.2	35.5 ± 0.1	35.8 ± 0.5
MGDG	36.0 ± 0.01	36.0 ± 0.02	36.0 ± 0.01	36.0 ± 0.01
DGDG	35.3 ± 0.04	35.6 ± 0.03	35.7 ± 0.01	35.6 ± 0.01
SQDG	33.7 ± 0.03	33.6 ± 0.02	34.0 ± 0.03	33.9 ± 0.02
PG	34.0 ± 0.02	34.0 ± 0.02	34.0 ± 0.01	34.0 ± 0.02

Note: ACL = (∑[*n* × % lipid])/100, *n* is the total number of carbons in the two fatty acid chains of each glycerolipid molecule. PC—phosphatidylcholines; PE—phosphatidylethanolamines; PG—phosphatidylglyceroles; PA—phosphatidic acids; DGDG—digalactosyldiglycerides, MGDG—monogalactosyldiglycerides, SQDG—sulfoquinovosyldiacylglyceroles. The table shows the average values and the standard deviations. The table shows the average values and the standard deviations. *—significant differences according to Student’s *t*-test.

**Figure 4 plants-12-00015-f004:**
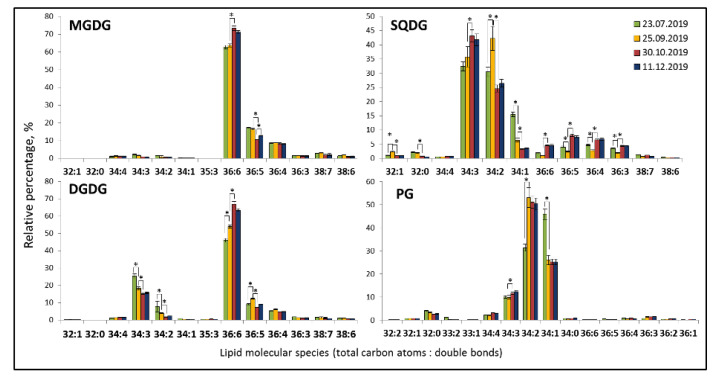
Seasonal changes in plastidic lipid molecular species in *Ephedra monosperma*. MGDG—monogalactosyldiacylglycerols; DGDG—digalactosyldiacylglycerols; SQDG—sulfoquinovosyldiacylglycerols; PG—phosphatidylglycerols. Bars represents standard deviations. *—significant differences according to Student’s *t*-test.

## 3. Discussion

The role of membrane lipids and their FAs in the adaptation of plants to low environmental temperatures, which may be of particular importance in the adaptation of evergreen plants to seasonal climate changes, is currently widely discussed. Evergreen plant species are capable of maintaining their assimilating apparatus for several years, while gradually replacing old assimilating organs with young ones [8], which is common for humid tropical and subtropical climates. Such species are characterized by an evolutionarily ancient type of rhythmic annual growth and development. The aridization and emergence of seasonal climate fluctuations caused the emergence of evergreen plants with a pronounced dormancy period in the annual cycle. In temperate and high latitudes, evergreens survive the unfavorable winter period [9] by rebuilding and preserving the photosynthetic apparatus. One of the mechanisms of adaptation to these conditions is the remodeling of lipid part of cell membrane, i.e., the reduction of some lipids and increase of others [2,10].

In these processes, the ratio of bilayer- and non-bilayer forming lipids is important. Bilayer lipids have a cylindrical shape and form a bilayer. They include PC, PI, PG, PS, DGDG. Non-bilayer lipids (e.g., PE, PA, and MGDG) with small headgroups and bulky acyl tails are cone-shaped and do not form bilayers. As a rule, bilayer lipids ensure membrane stability, whereas non-bilayer lipids are important for mediating proteolipid interactions and increasing the morphological plasticity of lipid bilayers [11]. In research devoted to the study of PL in plants and fungi exposed to various stressors, the PC/PE ratio is often used, which determines the structure and functionality of the membrane [12]. The cold hardening of *E. monosperma* was accompanied by a decrease in PC/PE ratio. The highest PC/PE value is characteristic of lamellar membrane structures, while a decrease in this ratio shows the formation of various inverted hexagonal structures in the membrane leading to membrane packing defects [12]. PC/PE ratio was shown to decrease during cold acclimation of plants [13,14] and fungi [15,16].

An increase in the local concentration of another non-bilayer lipid PA also leads to membrane packing defects, allowing the attachment of proteins that induce membrane fission/fusion. Local fluctuations in PA concentration accompany the formation of lipid droplets containing triacylglycerols. It is possible that sharp changes in PA concentration in *E. monosperma* are related to this process [17]. Interestingly, PA was the dominant lipid class of *E. monosperma*. It agrees quite well with report of Alqarawi et al. (2014) on the lipids of *Ephedra alata*, which contained one and a half times more PA compared to PC and PE under control conditions [18]. The large proportion of PAs in the phospholipid profile is characteristic of some extremophile organisms [19].

The FA composition of PA and its seasonal dynamics were similar to those of PC and different from PE. For example, 34:3 PE drastically increased during cold hardening, whereas 34:3 PC and PA did not change. This correlation of PA and PC molecular species’ composition suggests that two different pools of diacylglycerol are involved in the biosynthesis of PC (and PC-modified PA) and PE. PLDα1 is known to prefer PC as a substrate for PA production [20]. The increase of 36:4, 36:3 PA species under low temperatures allows to be identified as PA with 18-carbon chains at the sn-2 position synthesized in ER-, or the “eukaryotic” pathway. PA increase during cold acclimation reflects increased phospholipid synthesis in general associated with the autumnal reorganization of the membrane system of evergreens, where PA is the precursor metabolite for PL biosynthesis. PA also might transduce signals of environmental temperature change into the plant cells and trigger intracellular reactions. When the temperature decreased below 0 °C, the content of PA decreased rapidly (Figure 1). The inhibition of PA accumulation is a common and effective strategy for plants to tolerate freezing stress [21].

The major molecular species of PG in July were 34:1—46%, 34:2—32%. The occurrence of 18:1/16:0 and 18:1/16:1 in large amounts (78%) is characteristic of leaf PG of evergreen woody plants [22,23]. In addition, *E. monosperma* contains 34:3 (according to the data for woody plants 18:2/16:1)—10%, 34:4 (18:3/16:1)—3%, 32:0 (16:0/16:0)—4%. These major molecular species of PG are prokaryotic type synthesized in chloroplasts. Four PG molecular species of 16:0/16:0, 16:0/16:1t(Δ3-trans), 18:0/16:0, and 18:0/16:1t(Δ3-trans) referred to high-melting-point molecular species of PG (HMP-PG). Chilling-sensitive plants contain >30–60% HMP-PG of the total PG [24,25]. Sum of HMP-PG (32:0, 32:1 and 34:0, 18:0/16:11t(Δ3-trans) from the assimilating shoots of *E. monosperma* did not exceed 28% in July (Figure 4, which is characteristic of cold-resistant evergreens [23]. It should be noted that in PG 34:1 molecular species combination 18:1/16:0 is dominate over 18:0/16:1 for cold resistant evergreen plants [22,23]. The decrease of 34:1 from 45% to 25% and the concomitant increases of PG 34:2 from 31% to 52% were seen in PG by 25 September, when daily average air temperature dropped to 3.9 ± 2.9 °C. We suppose that 18:1 at the sn-1 position can be desaturated into C18:2 and then C18:3, while the C16:0 at this position generally remains unchanged (Figure 1 and Figure 4) similar to what has been observed in evergreen broadleaf relic shrub *Ammopiptanthus mongolicus* [26]. *A. mongolicus* growing in the central Asian desert can tolerate extremely cold weather (as low as approximately −30 °C during winters) and very dry climate like evergreen relic shrub *E. monosperma* in Central Yakutia.

It is known that plastid membranes consist of GLs, including MGDG, DGDG, and SQDG, as well as PG. DGDG/MGDG ratio is extremely important for plants because it determines bilayer formation, lipid-protein interactions, and interprotein interactions in thylakoid membranes. The latter refers to the energy interaction between the light-harvesting complex and PS II [27]. With the onset of persistent negative air temperature in October and November, the ratio of MGDG to DGDG significantly changed in favor of bilayer DGDG. Similar lipid remodeling has been reported in some plants [20,28,29] and it may be the result of the activity of SFR2 protein that transfers galactosyl residues from MGDG to various galactolipid acceptors, forming oligogalactolipids (including DGDG) and diacylglycerol, which is further converted to triacylglycerol [30]. Decline in the level of MGDG, a non-bilayer forming lipid, may contribute membrane stabilization under cold stress conditions.

Another function of MGDG is to interact with the pigments of the violaxanthin cycle, which protect against photooxidation [27]. Previously, Sofronova et al. [31] showed that in September, during the onset of low positive quenching temperatures in Central Yakutia, the amount of violaxanthin cycle pigments increased 1.7-fold in *E. monosperma* shoots, and their de-epoxidation level increased 3.6-fold. The amount of MGDG also increased under September temperature conditions, which is primarily related to the formation of chloroplasts and the development of their thylakoid system. It can be assumed that the maintenance of high levels of glycolipids in *E. monosperma* tissues in September contributes to the stability of the photosynthetic apparatus and the accumulation of assimilates during the transition to the quiescent state, and ultimately the survival of *E. monosperma* plants in the extreme conditions of the Yakutian cryolithozone.

In the winter period, PL dominated over GL, which is illustrated by the PL/GL ratio (Figure 2). Such a tendency has been reported earlier in some studies [28,32]. The decrease in glycolipid levels during the onset of subzero temperatures reflects the structural and functional reorganization of the photosynthetic apparatus, including winter ultrastructural changes in chloroplasts (the number of thylakoids in grana decreased and plastoglobules appeared) [33] and it is an integral part of the complex process of plant hardening. At the same time, phospholipids are supported on the level of cold-acclimated plants, ensuring the integrity of cell membranes during the forced resting phase.

During the winter period (from early November to mid-April), the snow cover eliminates the direct effect of solar insolation on plants and significantly mitigates the effects of low temperatures (Table 1). The penetration of solar radiation, depending on snow cover structure, is limited to the depth of 30–50 cm. However, its amount is small due to the great reflectivity of snow. Approximately 8% of all incomed to the snow cover surface radiation goes to the depth of 5 cm, 3–4%—10 cm, and 18–20 cm—1% [34]. It is noted that the height of snow coverage of plants in November is 5–7 cm, and only by the end of December reaches 10–15 cm. According to our data, during the winter months, when natural sunshine is 230–250 μmol/(m^2^ s), the illumination under the snow cover at a depth of 5, 10, and 15 cm is 20–25, 8–10 and 3–5 μmol/(m^2^ s), respectively. In March-April on sunny days, these values increased by 2.0–2.5 times. Consequently, unlike coniferous woody species [35], protective modifications both in pigment complex [31] and in the lipid metabolism of *E. monosperma* shoots wintering under snow cover play a special role in autumn during quenching, in spring after snowfall, and before the beginning of active shoot growth.

The modification of FA composition is considered to be one of the general defense systems against various biotic and abiotic stresses, including low-temperature [36]. An increase of the amount of unsaturated membrane lipids is a commonly observed response to low-temperature stress [20,37,38,39]. For *E. monosperma*, an increase in the degree of unsaturated lipids has been shown under cold hardening that is evident from the seasonal dynamics of the fatty acyl content of lipids (Table 2) and lipid profiling (Figure 3 and Figure 4, Table 3). The degree of acyl chain length (ACL) of most lipid classes was maintained during all seasons on the same level except for that of PE (Figure 4). The ACL of PE decreased during phase I of cold hardening and increased under negative temperatures.

## 4. Materials and Methods

Plant material and growth conditions. The plants of *Ephedra monosperma* J.G. Gmel. ex C.A. Mey. grow naturally in the botanical garden of the Institute of Biological Problems of Cryolithozone, Siberian Branch, Russian Academy of Sciences. The garden area is located on the second terrace above the flood plain of the Lena valley (62°15′ N, 129°37′ E). The objects of the study were shoots of *E. monosperma.* The samples were immediately fixed in liquid nitrogen and transported in Dewar vessels to the laboratory. For biochemical studies, the samples of the aerial parts of *E. monosperma* fixed in liquid nitrogen were dried in the lyophilizer (VirTis, New York, NY, USA). The experiments were carried out in 2019. Air temperature on the experimental plot were recorded with an accurancy ± 0.5 °C at 1 h intervals using a DS 1922L iButton thermograph (Dallas Semiconductor, Dallas, TX, USA). The average air temperature over growing season (May–September) were 13.1 and 13.9 °C, in 2019, respectively, and the total liquid precipitation amounted to 123 and 127 mm, respectively. The beginning of weak frosts was noted from the beginning of the second half of September, persistent lowering of night temperature below 0 °C level occurred after 30 September and 9 October, in 2019, respectively. Minimum air temperatures in winter were −48.4 °C. The snow cover was established on 13 October and 10 October, in 2019, respectively. The thickness of snow layer was 2–10 cm by the end of October, 15–21 cm in the middle of November, 18–26 cm in December, 24–34 cm in January.

For lipid extraction, a weighed portion of plant material (0.5 g) was fixed in liquid nitrogen and ground until a homogeneous mass was obtained. Cooled laboratory glassware and reagents were used. Then, 10 mL of a 1:2 chloroform/methanol mixture was added. Ionol was added to the mixture as an antioxidant (0.00125 g per 100 mL of the mixture). Everything was thoroughly mixed and left for 30 min until the complete diffusion of lipids into the solvent. The solution was transferred quantitatively into a separatory funnel through a filter. The mortar and filter were washed three times with the same solvent mixture. To separate the non-lipid components, water was added.

For the analysis of total lipids, the lower chloroform fraction was separated. Chloroform (high purity grade, stabilized with 0.005% amylene) was removed from the lipid extract under vacuum using the RVO-64 rotary evaporator (Mikrotechna, Praha, Czech Republic). To control the lipid extractability (%), the known amount of 10 µg of nonadecanoic acid (C19:0) was added at the homogenization stage. FAMEs were obtained using the Christie method [40]. Additional FAME purification was carried out by thin-layer chromatography (TLC) on glass plates with KSK silica gel (Reachem, Moscow, Russia). Benzene was used as the mobile phase. To visualize the FAME zone (Rf = 0.71–0.73), the plates were sprayed with 10% H_2_SO_4_ in MeOH and heated in an oven at 100 °C. The FAME zone was removed from the plate with a spatula and eluted from silica gel with *n*-hexane. The FAME analysis was performed by GLC using the 5973/6890N MSD/DS gas chromatograph–mass spectrometer (Agilent Technologies, Santa Clara, CA, USA). The detector was a quadrupole mass spectrometer. The ionization method was electron impact with an ionization energy of 70 eV. The analysis was performed in the mode of the total ion current recording.

An HP-INNOWAX capillary column (30 m × 250 μm × 0.50 μm) with a stationary phase (PEG) was used to separate the FAME mixture. The carrier gas was helium, and the gas flow rate was 1 mL/min. The evaporator temperature was 250 °C, the ion source temperature was 230 °C, and the detector temperature was 150 °C. The temperature of the line connecting the chromatograph with the mass spectrometer was 280 °C. The scanning range was 41–450 amu. The volume of the injected sample was 1 μL, and the flow divider was 5:1. The separation of the FAME mixture was carried out in isothermal mode at 200 °C. To identify FAs, the NIST 08 mass spectral library and the Christie FAME mass spectral archive were used [41]. The relative FA content was determined by the method of internal normalization in weight percent (wt.%) of the total content in the test sample, taking into account the FA response coefficient.

To characterize the degree of lipid unsaturation, the unsaturation coefficient (*k*) and the double-bond index (DBI) were used [42].
*k* = ∑Punsaturated/∑Psaturated,
DBI = ∑Pj nj/100,
where P is the acid content (%), Pj is the acid content (%), and nj is the number of double bonds in each acid.

The activity of acyl-lipid membrane ω9-, ω6-, and ω3-desaturases responsible for the introduction of double bonds into hydrocarbon chains of oleic (C18:1 (n-9)), linoleic (C18:2 (n-6)), and α-linolenic (C18:3 (n-3)) fatty acids was calculated as stearoyl (SDR), oleyl (ODR), and linoleyl (LDR) desaturase ratios [43] formulas as follows:SDR = (%C18:1)/(%C18:0 + %C18:1),
ODR = (%C18:2 + %C18:3)/(%C18:1 + %C18:2 + %C18:3),
LDR = (%C18:3)/(%C18:2 + %C18:3).

Individual classes of polar lipids were analyzed by two-dimensional HPTLC on silica gel plates 60 (10 × 10 cm) (Merck, Darmstadt, Germany) in a system of solvents: chloroform–methanol–water (65:25:4) in the first case and chloroform–acetone–methanol–acetic acid–water (50:20:10:10:5) in the second case [44].

The lipids were identified using standards for target components and specific reagents for individual functional groups [45].

The amounts of PL, GL, and sphingolipids were determined densitometrically using the Denskan (Lenkhrom, St. Petersburg, Russia). For this purpose, chromatograms were developed in 10% sulfuric acid in methanol followed by heating at 140 °C. The calculation of the content of individual classes of lipids in chromatograms was carried out using the DENS-14-12-03 program in linear approximation mode on calibration curves constructed using standard PC solutions (Larodan, Solna, Sweden), bovine cerebrosides, and MGDG (Sigma, St. Louis, MO, USA).

Statistical processing. The tables show the average data from three biological replicates and their standard deviations. The experimental data were statistically processed using the statistical analysis package in the Microsoft Office Excel 2017 environment. The statistical significance of the differences between the compared mean values was assessed using the *t*-test (*p* < 0.05).

### 4.1. Separation of Polar Lipid Classes by HPTLC

The separation of polar lipid classes was carried out by using a two-dimensional high-performance thin layer chromatography on silica gel 60 10 × 10 cm plates (Merck, Germany) in a solvent system of chloroform–methanol–water (65:25:4) in the first direction and chloroform–acetone–methanol–acetic acid–water (50:20:10:10:5) in the second direction [44]. After temporary visualization in iodine vapors lipid spots were scrapped from HPTLC plates and eluted with chloroform–methanol (1:1) at 4 °C overnight, after that, the solvent was removed by a rotary evaporator and samples were redissolved in 100% methanol. Polar lipid classes were identified by comparison of their retention times with those of standard samples.

### 4.2. Polar Lipid Analysis by Direct ESI-MS

The composition of molecular species of PC, PE, PA, PG, MGDG, DGDG, and SQDG was performed by high-resolution mass spectrometry using the MicrOTOF 10,223 time-of-flight mass spectrometer (Bruker Daltonics GmbH, Bremen, Germany) equipped with an electrospray ionization (ESI) source. The spectra were recorded during ionization via sputtering in an electric field in the positive ion mode during continuous direct probe injection in the range of 150–1000 *m*/*z* at a capillary voltage of 4 kV. The nebulizer gas pressure was 0.4 bar, and the drying gas flow was 4.0 L/min with temperature of 180 °C. The height of detected monoisotopic peak was used for quantification after isotope correction. PC, PE, MGDG, and DGDG species were identified as [M + Na]^+^ ions while PA, PG and SQDG species were identified as [M-H]^−^ ions. For detailed information about masses of identified molecular species see Appendix A online.

Although no corrections for varying mass spectral response to the various molecular species have been applied, these data provide for direct comparison of the relative amounts of each molecular species in samples compared with other samples.

Data were processed in the Compass Data Analysis Viewer 4.4 software (Bruker Daltonics). Identification of the individual molecular species was based on the *m*/*z* value of their monoisotopic adducts after isotope correction procedure. Peak annotation was carried out using LipidMaps. False positives were checked manually.

## 5. Conclusions

The acclimation of *E. monosperma* to low temperature was accompanied by the remodeling of membrane lipids. The increase in the non-bilayer lipids PE and DGDG, the increase in unsaturated C18:2(Δ9,12) and C18:3(Δ9,12,15) acids contribute to the stability of cell membranes and the survival of *E. monosperma* monocot plants in the extreme conditions of the Yakutian cryolithozone. Two types of temperature stresses, corresponding to the two phases of plant hardening, induced different biochemical modifications in lipid composition: low positive temperatures induced the increase of total PL content, the decrease of ACL in PE and the increase of glycolipid content, whereas under low negative temperatures the increase of ACL in PE and the decrease of glycolipid content, and the increase of DGDG/MGDG ratio, were observed. The current study lays the foundation for deeply analyzing the molecular mechanisms of plant tolerance to extremely low temperatures typical of natural conditions of permafrost ecosystems.

## Data Availability

Data are contained within the article.

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
