# Peer review of "Role of Lipids of the Evergreen Shrub Ephedra monosperma in Adaptation to Low Temperature in the Cryolithozone"

_plants, 2022, doi:10.3390/plants12010015_

Round 1

Reviewer 1 Report

This manuscript title “Role of lipids of the evergreen shrub Ephedra monosperma in adaptation to seasonal climate changes in the cryolithozone” analyzed the role of membrane glycerolipids during adaptation to the cold climate of the cryolithozone. The manuscript did not show substantial scientific observations or findings. The conclusion is redundant and should be condensed to the main results. Discussion is poor without any mechanistic approach. I suggest authors should revise the manuscript thoroughly.

Some minor comments:

1. DW: plant dry weight? But from the material section, the samples collected are fresh samples.

2. The plants grow naturally, there are other changing environmental factors that could affect the results (e.g. light, humidity, rainfall), this should be discussed.

3. Table 3, 4 are not marked the analysis of significance, there were no significant differences?

4. Line 131, line 202 both 2.2.

5. ** is used to mark significant differences in the graph, but is used to mark other concepts in the table. It is better to unify this and confused the readers.

6. Line 472: What part of the plant is it measured? This should add.

7. It's difficult to understand what is your own results and what is the previous results in discussion section.

Author Response

This manuscript title “Role of lipids of the evergreen shrub Ephedra monosperma in adaptation to seasonal climate changes in the cryolithozone” analyzed the role of membrane glycerolipids during adaptation to the cold climate of the cryolithozone. The manuscript did not show substantial scientific observations or findings. The conclusion is redundant and should be condensed to the main results. Discussion is poor without any mechanistic approach. I suggest authors should revise the manuscript thoroughly.

Answer: We thank the reviewer for valuable comments and advice.

Some minor comments:

  1. DW: plant dry weight? But from the material section, the samples collected are fresh samples.

Answer: Thank you for your comment. Added the following: The samples were immediately fixed in liquid nitrogen and transported in Dewar vessels to the laboratory. For biochemical studies, the samples of the aerial parts of E. monosperma fixed in liquid nitrogen were dried in the lyophilizer (VirTis, New York, USA).

  1. The plants grow naturally, there are other changing environmental factors that could affect the results (e.g. light, humidity, rainfall), this should be discussed.

Answer: The abiotic environmental factors in our experiments are typical for Central Yakutia and have changed little over the past decade. Short-term fluctuations with small amplitude in humidity, precipitation and photosynthetically active radiation (PAR), do not significantly affect on seasonal patterns of lipid composition.

  1. Table 3, 4 are not marked the analysis of significance, there were no significant differences?

Answer: Thank you for your comment. Added * to Table 3 and Table 4; *— significant differences according to Student’s t-test.

  1. Line 131, line 202 both 2.2.

Answer: Fixed, thanks for the note.

  1. ** is used to mark significant differences in the graph, but is used to mark other concepts in the table. It is better to unify this and confused the readers.

Answer: Yes you are right. We have changed the notation.

  1. Line 472: What part of the plant is it measured? This should add.

Answer: Thank you for your comment. Added «The objects of the study were shoots of E. monosperma».

  1. It's difficult to understand what is your own results and what is the previous results in discussion section.

Answer: Thank you for your comment. You're right. We have corrected the manuscript by shortening the results and discussion.

Reviewer 2 Report

In this work, Nokhsorob et al. investigated the role of lipids of Ephedra monosperma in adaptation to seasonal climate changes in the cryolithozone. This is an important addition to the existing knowledge which can contribute the lipid chemistry and its role in plant stress tolerance.

However, I am wondering why E. monosperma was used as a test plant. Do you have any background study on how it responds to climate change.

The title is misleading because there are many general and broad terms. How you will measure climate change. I suggest specifying these things.

Overall the manuscript is too wordy.

The results are too long. I strongly suggest concise this.

The discussion is also long.

Finish conclusion in a paragraph.

Author Response

In this work, Nokhsorob et al. investigated the role of lipids of Ephedra monosperma in adaptation to seasonal climate changes in the cryolithozone. This is an important addition to the existing knowledge which can contribute the lipid chemistry and its role in plant stress tolerance.

However, I am wondering why E. monosperma was used as a test plant. Do you have any background study on how it responds to climate change.

Answer: Thank you for your interest in our work.

Ephedra monosperma J.G. Gmel. ex C.A. Mey (division Gymnospermae, class Gnetopsida, subclass Pinida, family Ephedraceae Wettst., genus Ephedra) is   survivor from the Pre-Glacial Period and is one of the few evergreens shrubs grown in Central Yakutia. This species can be assigned to the group of light loving species adapted to growth in open well-lit areas. The habitats of E. monosperma are marked by severe continental arid climate characterized by extremely low winter (to –40…–45°Ð¡) and very high summer (up to 38°Ð¡) temperatures with an annual precipitation less than 200 mm, resulting in water deficit in air and soil. Its natural distribution areas are also characterized as sandy or calcareous rocks soils.  The extreme tolerance of this species to harsh environments makes it invaluable for exploring photoprotective and cold, drought tolerance mechanisms.

The dynamics of modulated chlorophyll fluorescence and carotenoid content in assimilating shoots of E. monosperma was analyzed during seasonal decrease in ambient temperature. Our experiments extended through all principal stages of frost hardening: the transition to dormancy that progressed first to “stage 1” hardening (at day/night temperatures of 4–10/0–5°C) and then to “stage 2” hardening (at temperatures from 0 to –10°C) as well as the winter dormancy (from –15 to – to –24°C). The lengths of light day shortened from 19.0 to 5.1 h. The average daily incidence of solar radiation decreased from 500…650 to 120…280 μmol/(m2s) until plants are covered with snow. The constitutive accumulation of zeaxanthin and the light screening secondary carotenoid rhodoxanthin was noted at near zero temperatures. The results suggest that principal photoprotective mechanisms during seasonal lowering of temperature are: (1) inactivation of PSII and dissipation of excitation energy in PSII reaction centers and (2) zeaxanthin_mediated energy dissipation in the antenna complexes. The first mechanism seems to prevail at early stages of seasonal cooling, whereas both mechanisms are recruited from the onset of sustained freezing temperatures. However, lipid profiling playing crucial roles in cold response of this species under low temperature have not been reported. In the present study, we firstly examined absolute and relative content and composition of fatty acids, total lipids, phospholipids, and galactolipids as principal constituents of cell membranes extracted from assimilating shoots of plants. Experiments were conducted in the period from July to the middle of December 2019.

The title is misleading because there are many general and broad terms. How you will measure climate change. I suggest specifying these things.

Answer: Thank you for your comment. Changed the title of the manuscript «Role of lipids of the evergreen shrub Ephedra monosperma in adaptation to low temperature in the cryolithozone»

Overall the manuscript is too wordy. 

Answer: You're right. We have corrected the manuscript by shortening the results and discussion.

The results are too long. I strongly suggest concise this.

Answer: You're right. We have corrected the manuscript by shortening the results.

The discussion is also long.

Answer: You're right. We have corrected the manuscript by shortening the discussion.

Finish conclusion in a paragraph.

Answer: You're right. The conclusion was made in one paragraph

We thank the reviewer for valuable comments and advice.
